# Exploring a combined biomarker for tuberculosis treatment response: protocol for a prospective observational cohort study

Frank Kloprogge [ID] ,[1] Ibrahim Abubakar [ID] ,[1] Hanif Esmail,[1,2] Vanessa Hack,[1] Heinke Kunst,[3] Timothy D McHugh,[4] Mahdad Noursadeghi,[5] Julian Surey,[1] Simon Tiberi,[3,6] Marc Lipman [ID] [7,8]

**Correspondence to**
Dr Frank Kloprogge;
f.kloprogge@ucl.ac.uk

## ABSTRACT

**Introduction** An improved understanding of factors explaining tuberculosis (TB) treatment response is urgently needed to help clinicians optimise and personalise treatment and assist scientists undertaking novel treatment regimen trials. Promising outcome proxy measures, including sputum bacillary load and host immune response, are widely reported with variable results. However, they have not been studied together in combination with antibiotic exposure. The aim of this observational cohort study is to investigate which antibiotic exposures correlate with sputum bacillary load and which with the host immune response. Subsequently, we will explore if these correlations can be used to inform a candidate combined biomarker predicting cure.

**Methods and analysis** All patients aged $\geq 18$, diagnosed with drug-sensitive pulmonary TB (culture or molecular test), eligible for standard anti-TB treatment, at selected London, UK TB Services, will be invited to participate in this observational cohort study (target sample size=210). Patients will be asked to give blood for host transcriptomics and antibiotic plasma exposure, in addition to standard of care sputum samples for bacillary load. Antibiotic plasma concentrations will be quantified using a validated liquid chromatograph triple quadrupole mass spectrometer (LC-MS/MS) assay and sputum bacillary load by mycobacterial growth incubator tube time to positivity. Expression from a total of 35 prespecified host blood genes will be quantified using NanoString®. Antibiotic exposure, sputum bacillary load and host blood transcriptomic time series data will be analysed using nonlinear mixed-effects models. Correlations between combinations of longitudinal biomarkers and microbiological cure at the end of treatment and remaining relapse free for 1 year thereafter will be analysed using logistic regression and Cox proportional hazard models.

**Ethics and dissemination** The observational cohort study has been approved by the UK's HRA REC (20/SW/0007). Written informed consent will be obtained. Results will be disseminated via publication, presentation and through engagement with institutes/companies developing novel anti-TB treatment combinations.

---

### Strengths and limitations of this study

► This study explores for the first time correlations between antibiotic exposure and both sputum tuberculosis (TB) bacillary load and host immune response.

► This enables correlations between different combinations of longitudinal markers of TB treatment response and TB treatment outcome to be explored.

► Such correlations can form the basis for a candidate biomarker that predicts cure from TB during the intensive phase of treatment.

► Sampling schedules are restricted to standard of care visits among a patient cohort treated for drug-sensitive pulmonary TB.

► Findings will need validation using data from patients with different forms of disease and in other geographical regions.

---

## INTRODUCTION

Globally, in 2019, an estimated 10.0 (9.0–11.0) million people fell ill with tuberculosis (TB). In the same year, approximately 1.2 (range: 1.1–1.3) million and 208 000 (range: 1 77,000–2 42, 000) HIV-negative and positive people died from TB.[1] The treatment of TB comprises a combination of antibiotics to optimise the effect and mitigate the emergence of resistance. It currently takes 6 months minimum to treat drug-sensitive TB (DS-TB) or multidrug-resistant (MDR) TB.[2 3] However, it has been shown that for 80% of patients, a shorter treatment duration suffices[4] although there are currently no biomarkers that can accurately identify these individuals. Such a biomarker could, for example, help identify patients, within the first 2 months of treatment, that need to continue after the intensive phase as they are not responding well. It could also help improve the design of prospective clinical

trials investigating rationally designed effective, shorter and/or individualised treatment options.

Several approaches, including measuring early bactericidal activity or host immune response, to predict favourable outcomes, have been tested.[5–7] Previous research shows that steeper bacterial kill-curves and bacterial kill-curves that are steep for longer correlate with microbiological cure at the end of treatment. Reoccurrence (ie, reinfection and relapse) was associated with less steep bacterial kill-curves.[8] These findings are consistent with work by others, which demonstrated that detectable *Mycobacterium tuberculosis* (*Mtb*) in sputum 2 and 3 months into treatment was associated with treatment outcome. However, the results also showed that culture conversion, as a predictor for treatment outcome, is not good enough for trial-level outcome predictions.[5] The predictive value of culture conversion is even less helpful for individual patients.[6]

Host immune response markers such as seen in the whole blood transcriptome can discriminate between active TB disease and healthy controls.[9] As the expression of certain gene signatures resolves during treatment of active TB,[10 11] this information could form the basis for identifying a biomarker of early response to treatment. This has been reported in patients with DS-TB, but early predictions (e.g., during week 1 and week 4 of treatment) of microbiological cure at the end of treatment were not accurate enough for clinical or trial application.[7]

Interactions between the longitudinal markers, for example, antibiotic exposure, sputum bacillary load and host blood transcriptome within the same patient have to date not been evaluated, even though a combination of antibiotic exposure, sputum bacillary load and host immune response would be expected to follow a composite pattern. The aim of this study is to explore a combined biomarker for response to treatment of TB. The primary objective is to investigate correlations between antibiotic exposure and sputum bacillary load and host immune response, with a secondary objective to explore trends between the aforementioned correlations and treatment outcome.

## METHODS AND ANALYSIS

This is a prospective, longitudinal and repeated-measure study that plans to recruit 210 drug-sensitive pulmonary patients with TB in London, UK. Patients will be followed during the 6-month treatment period at predefined visits. Formal follow-up visits are not usually scheduled during a 12-month post-treatment period, and instead patients are encouraged to contact the London TB Services immediately when signs and symptoms of possible TB disease reappear. Patients will be linked to the national TB database to enable monitoring for relapse or reinfection (table 1).

### Study site selection

TB notification numbers and rates display a steady decline in England since 2011. The majority of TB cases is reported in London (2018: n=1691 and rate=19.0/100,000) and other major cities such as Birmingham (West Midlands 2018: n=613 and rate=10.4/100,000). Coinfection with HIV is rare among patients with TB in England at 2.7% in 2018. Isoniazid and MDR/rifampicin resistant TB are also uncommon in England at 8% and 1.6% in 2018. Despite the most recent numbers being low in England,

| Table 1 Study design. | | | | | | | | | |
|---|---|---|---|---|---|---|---|---|---|
| D: day, W: week and M: month | D0 | D14 | W4 | W8 | W12 | M4 | M5 | M6 | Follow-up* |
| Consent | ✓ | | | | | | | | |
| Medical history | ✓ | | | | | | | | |
| Physical exam | ✓ | | | | | | | | |
| Bodyweight measure | | ✓ | ✓ | ✓ | ✓ | ✓ | ✓ | ✓ | ✓ |
| Laboratory test† | ✓ | ✓ | ✓ | ✓ | ✓ | ✓ | ✓ | ✓ | ✓ |
| Pregnancy test | ✓ | | | | | | | | |
| Sputum sampling‡ | ✓ | ✓ | ✓ | ✓ | ✓ | ✓ | ✓ | ✓ | ✓ |
| Blood plasma sampling | | ✓ | | ✓ | | | | ✓ | |
| Whole blood sampling | ✓ | ✓ | | ✓ | | | | ✓ | ✓§ |
| Medication record box¶ | ✓ | | | | | | | | |

*Passive follow-up phase, patients will be linked to the national TB database to enable monitoring of relapse or reinfection and are asked to contact the London TB Services when signs and symptoms of possible TB disease re-appear.
†Including full blood count, liver function tests (hepatic transaminases and total bilirubin), bio-chemistry (urea and electrolytes).
‡To be conducted until two consecutive negative samples; and sampling may be resumed when there are signs of treatment failure or relapse.
§To be performed only in case of a recurrence, that is, a positive sputum smear or culture after patient being successfully treated at month 6 (M6).
¶Medication record box, explain how to use it at the first visit.
TB, tuberculosis.

the prevalence of TB in London remains high compared with other major Western European cities.[12]

The relatively high prevalence of TB in London, in combination with well-resourced TB clinics, enables us to conduct this observational cohort. Serial sputum bacillary load quantification is common as part of standard of care, while drug-level measurements are now more frequent given their value in determining whether therapeutic drug levels have been achieved.[13] A limited amount of additional measures are, therefore, needed to enable us to answer the proposed research questions. These include: whole blood sampling for host blood transcriptomics and precise adherence monitoring using medication record boxes.

The London TB Services is organised into five geographical groupings. We intend to recruit from all London sectors. This will provide a diverse study population. The study started on 23 February 2021 and we expect to reach our target sample size in 24 months.

## Participant eligibility criteria and recruitment

Individuals attending the study clinics that meet all inclusion and none of the exclusion criteria are given patient information sheets. If the patient is willing to take part, then written consent will be obtained.

► Inclusion criteria
- Man or woman ≥18 years.
- Sputum sample suggesting *Mtb* based on culture or molecular test.
- Starting a pulmonary isoniazid and rifampicin susceptible *Mtb* treatment.
- Permission to be linked to (Public Health England) PHE national TB database.
- Willingness to attend study visits.
- Completed written consent.

► Exclusion criteria
- Pregnant or breast feeding.
- Social/medical conditions making study participation unsafe.
- Not eligible to start a course of DS-TB treatment.

Drug sensitivity testing at baseline is part of NHS standard care. This enables only patients with isoniazid and rifampicin susceptible *Mtb* infections to be enrolled.

## Assessment of objectives

In order to study both primary and secondary objectives, this study collects, in addition to clinical data obtained as part of standard of care, information on adherence to treatment and host blood transcriptomic analysis.

## Sample size assumptions

Given the primary objective, the sample size for this study was determined to power accurate and precise estimation of nonlinear mixed-effects model parameters that describe longitudinal pharmacological, microbiological and immunological data. At a sample size of 210, there is power to detect a correlation ≥0.19 between the different markers with 80% power and a 0.05 type 1 error rate.

The secondary objective aims to separate combined biomarkers between patients who had or had not reached microbiological cure at the end of treatment or had or had not finished treatment at 6 months, and patients who had or had no relapse during the first year after successfully finishing treatment. Assuming a 4.8% failure rate, defined as still on treatment after 12 months,[12] accounting for 10% drop out,[12] the study design is powered to detect a difference of 0.96 SDs in composite markers with 189 patients at 80% power with 0.05 type 1 error rate.

## Clinical covariates

Clinical covariates that are routinely collected as part of standard care will be collected from clinical notes and stored in a REDCap database.[14 15] These covariates include a medical history check, physical examination and pregnancy test at the D0 visit and at all consecutive visits bodyweight and laboratory tests including full blood count, liver function tests (hepatic transaminases and total bilirubin), urea and electrolytes (table 1).

## Evaluation of adherence

Patients taking part in this study will be given a medication record box to monitor the date and time that drugs have been taken.[16] Patients will be reminded by phone call or text about the clinic appointment the day before, if they have previously forgotten to bring their medication record boxes with them. This will be alongside standard NHS care, where patients who on initial or subsequent risk assessment are felt to be at high risk of poor adherence will be offered support with a range of options including directly observed therapy, video observed therapy and dosette boxes.

## Antibiotic plasma concentrations

Whole blood (0.5–1.0 mL) will be collected at ~2 hours postdrug intake on the D14, M2 and M6 visits (table 1). Plasma will be separated by centrifugation (at 1500 g for 10 min). Samples will subsequently be stored at −70°C, within 24 hours after collection, until shipment on dry ice for analysis.

Antibiotic plasma concentrations will be quantified using an Liquid Chromatograph Triple Quadrupole Mass Spectrometer (LC-MS/MS) as per standard of care protocols.

## Sputum bacillary load

In line with usual practice, patients will be asked to produce 2–3 sputum samples (2.5 mL) over a 24-hour period at the D0, D14 and M1 visit and then monthly until two negative consecutive sputum samples have been obtained. Where possible, sputum will be induced if patients stop coughing prior to two negative consecutive sputum samples have been obtained. Sputum bacterial load will be quantified using mycobacterial growth incubator tube time to positivity within 48 hours of production.

Isolates at baseline and subsequent treatment failure isolates will be subjected to whole genome sequence analysis.

### Host blood gene signature expressions

Whole blood (3 mL) will be collected directly into blood RNA tubes from patients at D0, D14, M2 and M6 visits for host blood transcriptomics. Samples will be stored in the freezer (−20°C) until analysis.

Expression of validated gene signatures, comprising 35 genes, will be quantified using NanoString®.[17 18]

### Modelling and statistics

A detailed analysis plan will be developed, but in brief, nonlinear mixed-effects models will be used for the primary objective. The first component comprises pharmacokinetic models that characterise how the drugs are absorbed in, distributed through, metabolised and eliminated from the body.[8] The second component will comprise a pharmacodynamic model that characterises clearance of bacillary load from sputum by the rate of elimination and the loss of steepness in bacillary killing over the course of treatment, that is, mono or biphasic decay.[8] The third component comprises nonlinear mixed-effects models that characterise resolution of gene expression for selected signatures during the course of treatment, which may be mono or biphasic.[9 17 18] All three model components consist of a structural model component and variability component that separates random variability between patients from residual variability. With these three types of models and a covariate modelling approach, drug levels—bacterial load correlations and drug levels—host immune response associations will be characterised.[19 20] Partial time series data, for example, patients switching treatment due to development of adverse events, will be retained for analysis.

A multivariate generalised logistic regression model will be used for the secondary objective, to explore a combined biomarker, comprising the aforementioned matrices, that best predicts microbiological cure at the end of treatment. The candidate combined biomarker's power to predict relapse during the 12 months following successful treatment will also be explored using a Cox proportional hazard model.

All aforementioned analyses will be performed using k-fold validation with the training partition of a proposed train:test split of 70:30 to obtain a more reliable predictive performance of the algorithm on the test split.[21] Furthermore, data from this study will need to be combined with that from other studies to further study the generalisability of the findings.

### Patient and public involvement

No patient and public involved.

### Project status

The Royal Free London NHS Foundation Trust will be the first site to recruit patients with subsequent roll-out to other London sites.

**Author affiliations**

[1]Institute for Global Health, University College London, London, UK

[2]Medical Research Council Clinical Trials Unit, University College London, London, UK

[3]Blizard Institute, Barts and The London School of Medicine and Dentistry, Queen Mary University of London, London, UK

[4]UCL Centre for Clinical Microbiology, Division of Infection & Immunity, University College London, London, UK

[5]Division of Infection and Immunity, University College London, London, London, UK

[6]Division of Infection, Royal London Hospital, Barts Health NHS Trust, London, UK

[7]Respiratory medicine, Royal Free London NHS Foundation Trust, London, UK

[8]UCL Respiratory, Division of Medicine, University College London, London, UK

**Contributors** FK and ML conceived the study and designed the protocol together with IA, HE, TDM and MN. FK wrote and IA, HE, VH, HK, TDM, MN, JS, ST and ML critically reviewed the manuscript.

**Funding** The study is funded through a Sir Henry Dale Fellowship jointly funded by the Wellcome Trust and the Royal Society (grant number 220587/Z/20/Z) and part of the medication record boxes was funded through a UKRI Medical Research Council fellowship (grant number MR/P014534/1), both awarded to FK.

**Competing interests** None declared.

**Patient and public involvement** Patients and/or the public were not involved in the design, or conduct, or reporting, or dissemination plans of this research.

**Patient consent for publication** Not required.

**Provenance and peer review** Not commissioned; externally peer reviewed.

**ORCID iDs**

Frank Kloprogge http://orcid.org/0000-0001-7213-4559

Ibrahim Abubakar http://orcid.org/0000-0002-0370-1430

Marc Lipman http://orcid.org/0000-0001-7501-4448

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
