## [Reviewer comments · BMJ Open]

ARTICLE DETAILS

TITLE (PROVISIONAL)	Exploring a combined biomarker for tuberculosis treatment response: protocol for a prospective observational cohort study
AUTHORS	Kloprogge, Frank; Abubakar, Ibrahim; Esmail, Hanif; Hack, Vanessa; Kunst, Heinke; McHugh, Timothy; Noursadeghi, Mahdad; Surey, Julian; Tiberi, Simon; Lipman, Marc

VERSION 1 – REVIEW

REVIEWER	Sathekge, M University of Pretoria, Nuclear Medicine
REVIEW RETURNED	25-May-2021

GENERAL COMMENTS	Exploring a combined bio-marker for tuberculosis treatment response; a prospective observational cohort study protocol In this manuscript, the authors present a comprehensive description of the protocol of a proposed study designed to investigate the role of combined blood antibiotic levels, sputum bacillary load and host immune response as a composite biomarker of response to treatment in patients with drug sensitive TB. The study seeks to investigate a problem of significant clinical importance especially since, up till today, there is still no biomarker predictive of response to anti-tuberculous therapy with acceptable accuracy. A few comments 1. Will routine drug sensitivity testing be done in all patients recruited into the study?2. Indicate if recruitment has commenced or not. Mention the date recruitment commenced or the date recruitment will begin if it has not begun already.
--

REVIEWER	Mirza, Shaper Lahore University of Management Sciences, Lahore, Pakistan, Biology
REVIEW RETURNED	30-May-2021

GENERAL COMMENTS	General Comments Line 46 “at end” should be replaced with “at the end” Specific comments Methodology: It is not clear how investigators will account for comorbid conditions, such as type 2-diabetes. Diabetes in particular poorly controlled
--

	diabetes and its associated metabolic syndrome results in immune impairments. Such impairments include alterations in numbers and function of macrophages and neutrophils, the two cells that drives a significant proportion of immunity to tuberculosis. Since measurements are dependent on immune responses also it might add a confounder in the study and should be recognized and included in the study plan. Screening for Diabetes should be part of the selection (inclusion or exclusion criteria) How will the investigator ensure adherence to treatment. Will the patient be administered medicine at the hospital/clinic/TB-camp
--	--

VERSION 1 – AUTHOR RESPONSE

Reviewer: 1

Dr. M Sathekge, University of Pretoria

Comments to the Author:

In this manuscript, the authors present a comprehensive description of the protocol of a proposed study designed to investigate the role of combined blood antibiotic levels, sputum bacillary load and host immune response as a composite biomarker of response to treatment in patients with drug sensitive TB. The study seeks to investigate a problem of significant clinical importance especially since, up till today, there is still no biomarker predictive of response to anti-tuberculous therapy with acceptable accuracy.

A few comments:

1. Will routine drug sensitivity testing be done in all patients recruited into the study?

Drug sensitivity testing at baseline is part of NHS routine care and will as such also be part of this study. An extra sentence below the inclusion and exclusion criteria has been added:

“Drug sensitivity testing at baseline is part of NHS standard care. This enables only patients with isoniazid and rifampicin susceptible M. tuberculosis infections to be enrolled.”

2. Indicate if recruitment has commenced or not. Mention the date recruitment commenced or the date recruitment will begin if it has not begun already.

This information has now been added.

Reviewer: 2

Dr. Shaper Mirza, Lahore University of Management Sciences, Lahore, Pakistan Comments to the Author:

Dear investigator, congratulation on conception of a valuable and timely study. There are few minor things that need your attention, otherwise, I found the study to be rigorous, and well planned.

*Line 46 “at end” should be replaced with “at the end”

Thank you for spotting, change made.

Specific comments

*Methodology: It is not clear how investigators will account for comorbid conditions, such as type 2-diabetes. Diabetes in particular poorly controlled diabetes and its associated metabolic syndrome results in immune impairments. Such impairments include alterations in numbers and function of macrophages and neutrophils, the two cells that drives a significant proportion of immunity to tuberculosis. Since measurements are dependent on immune responses also it might add a confounder in the study and should be recognized and included in the study plan. Screening for Diabetes should be part of the selection (inclusion or exclusion criteria).

We agree with the reviewer that comorbid conditions including HIV infection and type2-diabetes can substantially alter the host immune response. However, the correlations between antibiotic exposure, bacillary killing in sputum and host immune response have to date not been accurately quantified, let alone accounted for specific comorbidities. We believe, therefore, it would be inappropriate to make this an inclusion/exclusion criteria for the study. Nevertheless, we agree that this is an important area which needs to be investigated within a future prospective study. Within this project, we will be recording information on comorbidities that may impact on host response, and will look to adjust for them within our statistical data analysis.

*How will the investigator ensure adherence to treatment? Will the patient be administered medicine at the hospital/clinic/TB-camp?

Patients will be provided with electronic medication record boxes to enable detailed monitoring of treatment adherence. In line with standard NHS care, patients who on initial or subsequent risk assessment are felt to be at high risk of poor adherence will be offered support with a range of options including Directly Observed Therapy, Video Observed Therapy and dosette boxes. This has now been further explained under the "Evaluation of adherence" header.